# Modulation of Motor Awareness: A Transcranial Magnetic Stimulation Study in the Healthy Brain

**DOI:** 10.3390/brainsci13101422

**Published:** 2023-10-07

**Authors:** Adriana Salatino, Pietro Sarasso, Alessandro Piedimonte, Francesca Garbarini, Raffaella Ricci, Anna Berti

**Affiliations:** 1Department of Psychology, University of Turin, Via Po 14, 10123 Turin, Italy; 2NIT—Neuroscience Institute of Turin, Via Verdi, 8, 10124 Turin, Italy

**Keywords:** self-monitoring, motor awareness, premotor cortex, TMS, low frequency, awareness

## Abstract

Previous studies on the mechanisms underlying willed actions reported that the premotor cortex may be involved in the construction of motor awareness. However, its exact role is still under investigation. Here, we investigated the role of the dorsal premotor cortex (PMd) in motor awareness by modulating its activity applying inhibitory rTMS to PMd, before a specific motor awareness task (under three conditions: without stimulation, after rTMS and after Sham stimulation). During the task, subjects had to trace straight lines to a given target, receiving visual feedback of the line trajectories on a computer screen. Crucially, in most trials, the trajectories on the screen were deviated, and to produce straight lines, subjects had to correct their movements towards the opposite direction. After each trial, participants were asked to judge whether the line seen on the computer screen corresponded to the line actually drawn. Results show that participants in the No Stimulation condition did not recognize the perturbation until 14 degrees of deviation. Importantly, active, but not Sham, rTMS significantly modulated motor awareness, decreasing the amplitude of the angle at which participants became aware of the trajectory correction. These results suggest that PMd plays a crucial role in action self-monitoring.

## 1. Introduction

Although many of the processes underlying motor programming and execution are not accessible to consciousness, we are aware that we are moving (motor awareness) and that we desire to act (motor intention). Blakemore et al. [1] proposed that, for intentional movements, motor commands are selected and sent to the muscles to perform the action, while at the same time a prediction is made about the sensory consequences of the movement. This prediction (called the forward model) is based on an efference copy of the intended motor act and is compared, by a comparator system, to the actual feedback of the executed movement. According to this proposal, the forward model to be compared with the sensory feedbacks is the neural signal on which motor awareness is built [2,3]. Therefore, motor awareness seems to precede, rather than follow, the actual execution of an intentional action being, within certain limits, dissociated from it.

In a seminal experiment, Libet demonstrated that subjects become aware of a hand movement before the actual onset of muscle contraction [4], whereas Haggard and Magno [5] found that interfering, through single-pulse TMS, with the activity of the left primary motor cortex (M1) resulted in a significant delay of right-hand movements but had little effect on the time the subjects perceived the movement (assessed by asking participants to indicate the position of a rotating clock hand). Conversely, single-pulse TMS of the anterior frontal areas (with the coil placed at the standard FCz site) resulted in smaller delays in actual Reaction Times (RTs) but larger delays in the assessment of the timing of manual response, an ability related to motor awareness. This shows that motor awareness does not co-vary with the experimentally induced delay in motor response. In other words, once the intention to perform an action is formed, the motor response may be delayed, but the motor awareness, already triggered by the intentional stance, is not affected.

Motor awareness also can be reported in absence of any intentional movement. Indeed, brain-damaged patients with anosognosia for hemiplegia (i.e., patients who deny their paralysis) subjectively report the feeling of having performed an action with the paralysed limb [6,7]. This phenomenal experience has its measurable counterpart in the fact that the pretended action with the paralysed hand actually affects the spatiotemporal parameters of the movements of the healthy hand [8,9,10,11,12,13]. Interestingly, on the basis of lesional data in anosognosic patients, Berti and colleagues [6,14] proposed that the right premotor cortex (PM, especially Area 6) is part of a neural circuit for motor monitoring, and thus contributes to the operation of one of the comparator systems described by Blakemore et al. [1] (see also Haggard, 2005 [15]). In particular, previous studies have suggested the involvement of the premotor [6,16,17] and insular cortices [6,16,18] for the process of conscious motor monitoring, the basal ganglia, insulo-frontal, temporal and parietal structures for explicit and implicit motor awareness [19], and mesial–frontal [5,20,21] and posterior–parietal areas [22] for the intentional component of the motor act.

More recently, non-invasive brain stimulation evidence [23] has shown that cathodal transcranial Direct Current Stimulation (tDCS) of the PM, but not of the Posterior Parietal Cortex (PPC), affects the subjects’ self-confidence about their contralateral (left) hand motor performance, consistent with the idea of a role of right PM in the conscious monitoring of voluntary motor acts. On the contrary, tDCS over PM does not interfere with monitoring of involuntary muscle contractions induced by TMS over the hand motor area [24].

Taken together, previous findings suggest a role of right PM in the conscious control of voluntary action, but they do not provide direct evidence of its causal involvement. In the present study, we used a task in which participants were asked to draw a straight line either with the left hand (i.e., the hand contralateral to the stimulated side) or with the right (ipsilateral) hand. During the execution of the requested movements, the visual feedback of subjects’ actual motor performance was experimentally deviated from the real trajectory in most of the trials to create a mismatch between the movement they executed and the movements they viewed on a computer screen. This mismatch led the subjects to correct their trajectory in the opposite direction in order to draw a straight line. Previous studies (e.g., [25,26,27,28]) showed that, within certain limits of deviation, subjects did not become aware of the modified trajectory they performed. In other words, in the manipulated trial, until certain degrees of deviation, subjects still believed they were tracing a straight line, as required by the task. Therefore, in this experimental setting, subjects performed (erroneous) movements they were not aware of. In Fourneret and Jeannerod’s (1998) interpretation, this finding demonstrated that the subjects became aware of the movement they intended to perform (a straight movement) rather than the movement they actually performed (a deviated movement). This is, again, consistent with the idea that motor awareness is mainly constructed on a predictive code and not exclusively on the actual sensory feedback.

In the present study, we aimed at exploring the role of the right PM in motor monitoring by means of repetitive Transcranial Magnetic Stimulation (rTMS), which allows drawing causal links between the stimulated brain regions and the observed behaviours [29,30]. In our experiment, indeed, in order to interfere with motor awareness, we applied, before the execution of the task, 1 Hz rTMS over the right PMd cortex [31]. The task was also performed after Sham rTMS stimulation of the same area.

Our first prediction was that if the PM plays a pivotal role in motor awareness, then interfering with its activity using rTMS, when the subjects perform their “deviated” trajectories, further affects their action monitoring.

However, it is worth noting that while an inhibitory TMS usually worsens subjects’ responses, by decreasing the activity of the targeted areas [32,33,34], a few studies unexpectedly showed that inhibitory rTMS of the right PMd could enhance subjects’ performance [35,36,37]. Consequently, if rTMS has an inhibitory effect on motor monitoring, we should expect an increase in the angle at which subjects become aware of the deviated trajectory (i.e., a decrease in motor awareness), whereas if rTMS has an enhancing effect on motor monitoring, we should expect a decrease in the angle at which subjects become aware of the deviated trajectory (i.e., an increase in motor awareness).

As for the side of the body where right rTMS may have an effect, if the right PMd has a control only over the contralateral hand, we should expect a modulation of motor monitoring only for the left-hand action. However, if the right PM controls the awareness of both hand movements, we should expect to find a modulation of motor monitoring for both hands. Finally, we expect to observe a modulation of motor awareness in the active but not in the Sham rTMS condition.

Crucially, and independently from the outcome of the stimulation, a modulation of rTMS on subjects’ capability of detecting action deviation would be another fundamental step for demonstrating the key role of the PMd in the construction of motor awareness.

## 2. Materials and Methods

### 2.1. Participants

Fourteen healthy right-handed healthy volunteers (11 women, 3 men, age: 21–57 years, mean age 25.8; Standard Error—SE = 2.5) participated in the study. We based the choice of our simple size on previous TMS studies on motor cognition [5,31,38] and/or cognitive studies using 1 Hz rTMS [32,33]. Participants were selected according to the TMS exclusion criteria [39] and they provided their informed consent to participate in the study, previously approved by local Ethics Committee of the University of Turin (number of protocol: 24001). The study was conducted in accordance with the Declaration of Helsinki for human participants. The Edinburgh Handedness Inventory-revised [40] test was administered to ensure that all subjects were actually right-handed.

### 2.2. Set Up

The experimental set up consisted of a 30 × 40 cm graphic tablet placed in a wooden box on a desk and connected to the computer. The computer was also connected to an LCD screen placed on top of the wooden box, 30 cm above the graphic tablet. A hole in the wooden box allowed the participant insertion of one of the hands inside the box, thus excluding it from the subjects’ view (See Figure 1). The subject was seated on a comfortable fixed chair in front of the desk (both the graphic tablet and the screen were aligned with the subject’s trunk midline) and they could only see the screen below the chin. The chair was fixed so that the distance between the subject and the desk was kept equal across conditions (i.e., 60 cm). The hand inside the box held a pen stylus, while the other hand simply rested on the participant’s leg.

### 2.3. Procedures

The task was to trace a straight line from the starting position to the target position. During task execution, while tracing the vertical line on the tablet, the subjects only saw the line appearing on the computer screen. In the Artificially Deviated (AD) trials, the output of the graphic tablet was processed by the computer using a simple algorithm that added a constant linear directional bias to the right or left of varying amplitude, so that the trajectory drawn by the subject appeared displaced to the left or right according to an angle defined by the bias (i.e., Left Deviation—LD from −1° to −25° and Right Deviation—RD, from +1° to +25°). The trajectory on the screen was the only visual feedback of the actual movement available to the subject and, after the target was reached, they were asked to indicate with “yes” or “no” whether the trajectory they saw on the screen corresponded to the movement actually performed or not (see Figure 1). Subjects performed 51 trials in each experimental condition (i.e., No Stimulation, rTMS and Sham). There was also one trial with a deviation of 0°, indicating a perfect coherence between the visual feedback and the actual movement. Participants performed the task in three different conditions: No Stimulation (NS), after real 1 Hz rTMS (rTMS), delivered to the right PMd (900 pulses, at 90% of resting Motor Threshold, rMT), after Sham stimulation (SHAM), with the coil held perpendicularly to the right PMd. For each condition, participants performed the task with the right (RH) and the left (LH) hand. NS, active and Sham stimulation measurements were collected on 3 different days with an interval of about one month between sessions in order to prevent potential learning. The order of the 3 conditions, as well as the order of the hand, was randomised between participants. Within participants, the order of the hand used for the task was maintained across sessions.

### 2.4. Transcranial Magnetic Stimulation

In the rTMS and Sham conditions, the subject performed the task soon after 15 min of real repetitive TMS or Sham stimulation, respectively. In the Sham stimulation, the coil was positioned over the same area as in the real stimulation, but it was held in a position perpendicular to the subject’s scalp. RTMS was performed with a Magstim Rapid2 system with a focal coil (70 mm figure-of-eight). The participants’ resting Motor Threshold (rMT) was defined as the lowest pulse intensity able to elicit a visible twitch in the abductor pollicis brevis muscle of the right hand in at least five of ten consecutive stimulations of the motor hotspot [34]. The average resting Motor Threshold was 53.4 (SD = 5.38) of maximum machine output. Then, fifteen minutes of low-frequency rTMS (900 pulses, 1 Hz at 90% of rMT) was delivered over the right dorsal premotor cortex (PMd), defined as 2 cm anterior and 1 cm medial to the previously defined M1 hotspot [41]. Soon after the end of the stimulation, participants were asked to look at the screen in order to start the experimental task.

### 2.5. Data Analysis

Statistical analyses were conducted using Statistica 6.0. Only data from trials with manipulated angles were used for the analysis, so we excluded the trials without deviation (i.e., the 0). In order to establish at which degree each subject became fully aware of the deviation, the angle at which the subjects recognised the presence of a mismatch between what they saw on the screen and what they had actually traced was recorded and analysed. The dependent variable for the data analysis was therefore the angle of deviation (toward the left and/or the right side) at which the subject started to consistently answer “no” (e.g., at the 14 degree), for at least two consecutive degrees. On the degree selected using the above criteria, a 3 × 2 × 2 ANOVA with three within-subject factors, COND (No Stimulation—NS, rTMS, and Sham), SIDE (Right and Left DEVIATION), and HAND (left and right hands) was used to directly test the differential effects of NS vs. Real and vs. Sham rTMS.

## 3. Results

Analyses reveal that the factor COND was significant [F (2, 50) = 7.3; *p* = 0.001; partial η^2^ = 0.228]. Crucially, post hoc analyses (Duncan’s test) showed that participants became aware of their deviation at significantly smaller angle soon after the rTMS (*p* = 0.0006, mean = 11.09, SE = 0.9) with respect to the NS (mean = 14.34, SE = 1.1) and the Sham conditions (*p* = 0.01, mean = 13.3, SE = 1.1), independently from the hand used to perform the task and from the direction of the deviation (see Figure 2). This result shows that rTMS facilitated awareness compared to the other conditions.

Results also showed a significant three-way interaction between COND X SIDE X HAND [F (2, 50) = 3.79; *p* = 0.02; partial η^2^ = 0.228]. Post hoc analysis (Duncan’s test) revealed that when the task was performed with the left hand, subjects were significantly more aware of their own performance soon after rTMS (*p* = 0.003, mean = 11.53, SE = 1.6) than BS (mean = 16, SE = 1) and Sham (*p* = 0.008, mean = 15.46, SE = 1.41) in the Left Deviated trials. Conversely, when subjects performed the task with the right hand, they were significantly more aware of the deviation in the rTMS condition (mean = 11.28, SE = 1.46) compared to the BS (*p* = 0.0009, mean = 16.35, SE = 1.19) and the Sham (*p* = 0.01, mean = 14.92, SE = 1.3) in the Right Deviated trials (see Figure 3).

Taken together, these results suggest that rTMS of PMd significantly affected participants conscious self-monitoring, and that the deviation direction, as well as the hand used to perform the task, influenced subjects’ action self-monitoring.

## 4. Discussion

In the present study, we investigated the role of the right PMd in action self-monitoring in healthy volunteers by evaluating motor awareness modulation through the application of inhibitory rTMS. To obtain a behavioural measure of motor awareness, we referred to the well-known paradigm of Fourneret and Jeannerod where healthy volunteers are requested to judge their actions in a task in which self-generated movements are experimentally deviated [25,26,27,28].

More specifically, we investigated whether the application of low-frequency rTMS over the right PMd may affect the monitoring of trajectory deviations. Consistent with our expectations, we found a modulation of subjects’ motor awareness when movement perturbation was at 11 degrees of deviation. Interestingly, although the results reported in previous studies usually show worsening performance when inhibitory rTMS is applied [34], we found that inhibitory rTMS over the right PMd improved participants’ motor awareness (i.e., it decreased the angle at which they became aware of the deviation). This is in line with some results reported in previous studies showing facilitatory effects of the low-frequency rTMS protocol [35,36,37].

One possible explanation for the observed facilitation following 1Hz rTMS protocol is that it may be due to the phenomenon known as paradoxical facilitation (PF), for which behavioral facilitation may result from disruption or inhibition of brain activity [42,43,44]. PF was first described in brain-damaged patients who performed better than normal subjects on specific tasks [45,46]. Recently, it has been reported that PF can be induced by low-frequency rTMS in healthy participants. For example, Buetefisch and co-workers [47] showed that participants’ accuracy on a task that required a higher level of precision for both hands increased after low-frequency rTMS applied to the left M1. In a previous study, Avanzino et al. [48] demonstrated an improvement of ipsilateral motor accuracy following 1 Hz rTMS over M1 that lasted the period of stimulation up to 30 min. Similarly, Schwarzkopf et al. [49] demonstrated that low-intensity TMS over the visual cortex facilitated the detection of weak motion signals, while higher intensities resulted in impaired detection of stronger motion signals (see also Pascual-Leone et al. (2012) for a review [50]). Therefore, in healthy subjects the effects of low-frequency rTMS on the brain can either worsen [34] or improve subjects’ performance [35,36,37]. However, the specific mechanisms of how non-invasive brain stimulation induces PF in healthy individuals are not yet fully understood. One of the recently proposed explanations is the stochastic resonance model, which postulates that introducing small amounts of noise into a system may promote low-level signals, which, in turn, enhance functions within that system. Whatever the explanation, the crucial finding of our study is the modulation of conscious experience obtained by delivering rTMS over the PMd. This demonstrates the causal relationship between the premotor cortex and motor awareness, confirming that the premotor cortex can be considered an important hub of the circuit related to the construction of the conscious experience of actions.

As for the side of the body controlled by the right PM, our results show that stimulation of the right PMd affects motor awareness for both, the contralateral (left) and ipsilateral (right) hands. These findings showing a right-hemispheric control of both hands suggest a right hemispheric specialization for motor monitoring mechanisms. It is worth noting, however, that studies coming from different experimental paradigms have suggested that also the left pre-motor cortex seems to be involved in motor action monitoring. For instance, there are a few cases of anosognosia for the right hemiplegia (that is, anosognosia following left, instead of right, brain damage [51]) while Fornia et al. [52] found in an experiment with awake surgery that Direct Electrical Stimulation (DES) of PMC dramatically altered the patients’ motor awareness, making them unconscious of the motor arrest induced by the same stimulation. Given all these results, we might suggest that while the right hemisphere may have a control for motor action executed by both hands, the left hemisphere may have a control only on the right hand. This proposal is reminiscent of one of the theories put forward to explain the deployment of attention in space and the data on neglect [34,53]. Further investigation should consider this possibility.

Finally, we also found an unexpected result: an effect on participants’ awareness of the deviation direction related to the hand used to perform the task. Indeed, in the No Stimulation condition, participants were less aware of deviating from a straight trajectory in the Left Deviated trials when the task was performed with the left hand, and in the Right Deviated trials when the task was performed with the right hand. We can speculate on this result by referring to the well-known Simon effect, an attentional effect described as a stimulus-induced bias in response selection [54], in which manual responses to a visual stimulus are facilitated when there is congruence between the stimulus side and the responding hand (stimulus response compatibility effect [55]). In our experimental setting, a condition of hand-space compatibility was realised when the subject had to perform the task with the right (or left) hand and the line that was projected on the screen deviated towards the right (or left) space. The hand-space contingency created by the AD facilitated and enhanced attention to that space by interfering with the generation of awareness of the movements toward the opposite space that the subjects had to perform in order to correct the trajectories (see Freud et al. 2015 for the presence of the Simon effect in the motor trajectory task [56]).

## 5. Limitations of the Study

The present study has three main limitations. First, we did not perform stimulation of the left PMd, which prevents formulation of any definite conclusion about the role of the right PMd on motor awareness. Considering that the left pre-motor cortex also seems to be involved in action control [52], it will be crucial to investigate the effect of the stimulation of the left PMd on motor awareness. Second, in our task, we did not investigate the role of the right Parietal Cortex, which, according to some authors, is involved in motor intention [23]. Therefore, further studies targeting the Parietal Cortex are needed to clarify the role of different brain areas in action monitoring. Finally, it is possible that the medium effect size of the present study is due to the limited sample size (n = 14), although it is quite similar to the sample of previous TMS studies (see, for example, [32,34,38]).

## 6. Conclusions

Our results, showing a modulation of motor awareness by the application of rTMS to the right PMd, demonstrated that this region plays a crucial role in action self-monitoring. Although the interference with its activity improved the subjects’ motor awareness, our study suggests that one of the comparator mechanisms proposed by the Blakemore et al. [1] model, responsible for the conscious monitoring of motor acts, is located in the right PM [6]. Given the functional enhancement effect that we found when rTMS was administered to the PMd, it is worth considering this procedure as a possible treatment for motor awareness disorders. As already pointed out, a limitation of our study is that we did not test the effect of rTMS over the left PMC. This would be crucial to draw firm conclusions about the different involvement of the two hemispheres in action self-monitoring. Therefore, further investigations, also targeting different areas and increasing the numbers of participants, both in the right and left hemisphere, are needed to clarify the different components of the motor monitoring circuit and their specific role in generating the conscious experience of action.

## Figures and Tables

**Figure 1 brainsci-13-01422-f001:**
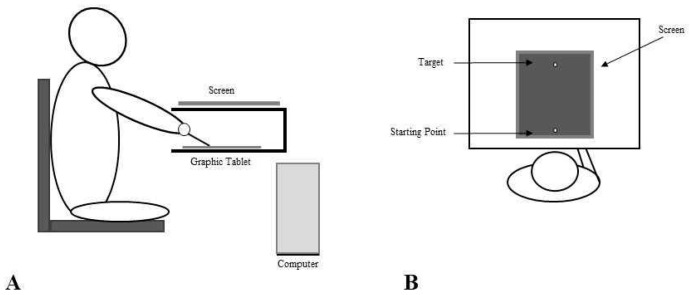
Experimental set up and procedure. The experimental set up consisted of a 30 × 40 cm graphic tablet placed in a wooden box on a desk and connected to the computer. An LCD screen was placed on top of the wooden box. A hole in the wooden box allowed the participants insertion of one of the hands inside the box, so that it could not be seen. The subjects were seated on a comfortable fixed chair in front of the desk (both the graphic tablet and the screen were aligned to the subjects’ trunk midline) and they could only see the screen below the chin (Panel **A**). The subjects were instructed to reach, with the pen tip, a yellow target (4 × 4 mm) located on the sagittal axis at 22 cm from the starting point, by drawing a continuous line as fast as possible. After the target was reached, they were asked to indicate with “yes” or “no” whether the trajectory they saw on the screen corresponded to the movement actually performed or not. Subjects performed 51 trials in each experimental condition (i.e., No Stimulation, rTMS and Sham). For each trial, the software randomly applied the trajectory deviation, which ranged from 25° to the left (LD, i.e., −25° from the 0, with negative values indicating a leftward perturbation) to 25° on the right (RD, i.e., +25° from the 0), with a trial for each degree of deviation. There was also one trial with a deviation of 0°, indicating a perfect coherence between the visual feedback and the actual movement (Panel **B**).

**Figure 2 brainsci-13-01422-f002:**
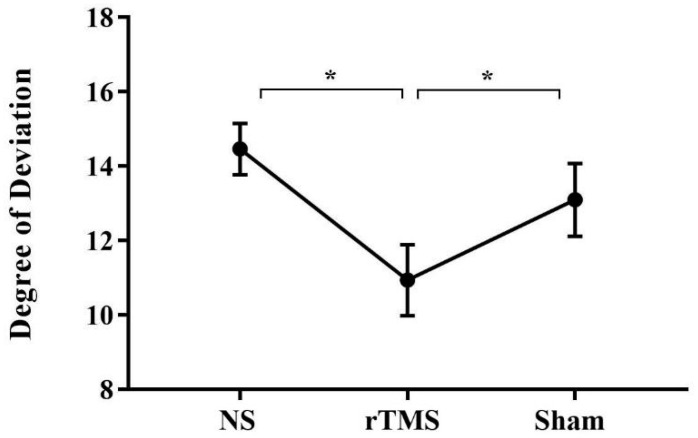
Mean degree at which subjects became aware of the artificial deviation in the three conditions. Mean degree at which subjects became aware of the artificial deviation in the BS (mean = 14.34, SE = 1.1), soon after the rTMS (mean = 11.09, SE = 0.9) and in the Sham conditions (mean = 13.3, SE = 1.1). Error bars represent standard error of means; *, significant. NS = No Stimulation, rTMS = repetitive Transcranial Magnetic Stimulation, Sham = Sham stimulation.

**Figure 3 brainsci-13-01422-f003:**
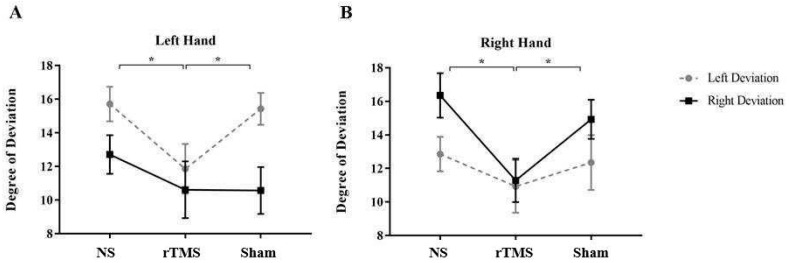
Mean degree at which subjects became aware of the artificial deviation in the three conditions and with the two deviations. Mean degree at which subjects became aware of the artificial deviation in the three conditions and with the two deviations when the task was performed with the left hand (Panel **A**) and with the right hand (Panel **B**). Error bars represent standard error of means; *, significant. NS = No Stimulation, rTMS = repetitive Transcranial Magnetic Stimulation, Sham = Sham stimulation. Taken together, these results indicate that rTMS of PMd significantly affected participants’ conscious self-monitoring, and that the deviation direction, as well as the hand used to perform the task, influenced subjects’ action self-monitoring.

## Data Availability

The data presented in this study are available on request from the corresponding authors.

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
