# Peer review of "Modulation of Motor Awareness: A Transcranial Magnetic Stimulation Study in the Healthy Brain"

_brainsci, 2023, doi:10.3390/brainsci13101422_

Round 1

Reviewer 1 Report

Comments and Suggestions for Authors

The authors of the manuscript, ' Modulation of motor awareness: a transcranial magnetic stimulation study in the healthy brain' have tried to investigate the role and relevance of dorsal premotor cortex (PMd) in motor awareness by modulating its activity by applying inhibitory rTMS to PMd before a specific motor awareness task.  The paper is well written but some points need to be corrected/rectified for its better exposition. These are

1.         This study suffers from low statistical power because of a very small sample size (n=14). It may inflate effect size and infuse erroneous results because of false negative errors. You may see the importance of sample size for reproducibility from the perspective of tDCS effects (PMID: 27679568). Even a small sample size in such a simulation study was called a research area of bad science (PMID: 26652115). Authors must give either an explanation or rectify it by increasing the sample size.

2.         Figure 1 is not visually comprehensible and will not represent what the authors want to show. I suggest authors may put a real photograph of the experimental setup of their laboratory.

3.         The full form of the abbreviation must be written where it comes first time in the text. For instance, abbreviations of PPS (line 65), and BS (line 202) are missing.

4.         The cited references such as [28,29,30,31,32] should be written as [28-32]. Check the whole text for it. Lines 97, 98, 239, 251, 264

5.         The authors have mentioned the aim of the study in lines 71-73, which should be written in the last paragraph of the introduction section.

Comments on the Quality of English Language

Minor English editings are required.

Author Response

For research article: Modulation of Motor Awareness: a Transcranial Magnetic Stimulation Study in the Healthy Brain

Response to Reviewer 1 Comments

1. Summary

The authors of the manuscript, ' Modulation of motor awareness: a transcranial magnetic stimulation study in the healthy brain' have tried to investigate the role and relevance of dorsal premotor cortex (PMd) in motor awareness by modulating its activity by applying inhibitory rTMS to PMd before a specific motor awareness task.  The paper is well written but some points need to be corrected/rectified for its better exposition. These are

2. Questions for General Evaluation

Reviewer’s Evaluation

Response and Revisions

Does the introduction provide sufficient background and include all relevant references?

Yes

Are all the cited references relevant to the research?

Can be improved

Thank you, we have added additional reference to the manuscript

Is the research design appropriate?

Can be improved

Are the methods adequately described?

Can be improved

Are the results clearly presented?

Can be improved

Are the conclusions supported by the results?

Can be improved

3. Point-by-point response to Comments and Suggestions for Authors

Comments 1:

This study suffers from low statistical power because of a very small sample size (n=14). It may inflate effect size and infuse erroneous results because of false negative errors. You may see the importance of sample size for reproducibility from the perspective of tDCS effects (PMID: 27679568). Even a small sample size in such a simulation study was called a research area of bad science (PMID: 26652115). Authors must give either an explanation or rectify it by increasing the sample size.     

Response 1: We thank the reviewer for raising this critical issue. We based the choice of our simple size on previous TMS studies on motor cognition (N=7, Haggard and Magno, 1999 Exp Brain Res; N= 10, Christensen et al., 2010, PLoS ONE; N= 16, Pyasik et al., 2019, Social cognitive and affective neuroscience) and/or cognitive studies using1 Hz rTMS (N= 15 per group, Ando’ et al., 2015, Brain Res; N= 13, e.g. Salatino et al., 2019, Front Psych). We have now added this point in the methodology section as also requested by Reviewer 2 (section: Materials and Methods, page 3, line: 127-130) and discussed this point in the limitation section (page 9, line 330-333) . However, we agree with the reviewer that sample size for tDCS studies need to be larger as discussed in the papers suggested by the reviewer.     

Comments 2: Figure 1 is not visually comprehensible and will not represent what the authors want to show. I suggest authors may put a real photograph of the experimental setup of their laboratory.

Response 2: Unfortunately, a real photograph of the experimental setup is not available, as the study was conducted in a lab that, similarly to other labs in Turin, were recently moved to another location and the original set up dismounted. When drawing this figure, we have referred to the original by Fourneret and Jeannerod (1998) and Slachevsky et al. (2001), as we have used a very similar setup.

Comments 3: The full form of the abbreviation must be written where it comes first time in the text. For instance, abbreviations of PPS (line 65), and BS (line 202) are missing.

Response 3: Thank you for pointing this out. We have added full form of the abbreviation when they are missing.

Comments 4: The cited references such as [28,29,30,31,32] should be written as [28-32]. Check the whole text for it. Lines 97, 98, 239, 251, 264.

Response 4: Thank you for pointing this out. We have now formatted the references as requested.

Comments 5: The authors have mentioned the aim of the study in lines 71-73, which should be written in the last paragraph of the introduction section.

Response 5: We thank the reviewer for his/her observation. We have revised the section as requested, and we wrote the aim of the study in lines 92-95.

4. Response to Comments on the Quality of English Language

Point 1: Minor English editings are required.

Response Point 1: We thank the reviewer for his/her observation. We have revised the English in the current version.

5. Additional clarifications

For review article

Response to Reviewer X Comments

1. Summary

Thank you very much for taking the time to review this manuscript. Please find the detailed responses below and the corresponding revisions/corrections highlighted/in track changes in the re-submitted files. [This is only a recommended summary. Please feel free to adjust it. We do suggest maintaining a neutral tone and thanking the reviewers for their contribution although the comments may be negative or off-target. If you disagree with the reviewer's comments please include any concerns you may have in the letter to the Academic Editor.]

2. Questions for General Evaluation

Reviewer’s Evaluation

Response and Revisions

Is the work a significant contribution to the field?

[Please give your response if necessary. Or you can also give your corresponding response in the point-by-point response letter. The same as below]

Is the work well organized and comprehensively described?

Is the work scientifically sound and not misleading?

Are there appropriate and adequate references to related and previous work? 

Is the English used correct and readable?       

3. Point-by-point response to Comments and Suggestions for Authors

Comments 1: [Paste the full reviewer comment here.]

Response 1: [Type your response here and mark your revisions in red] Thank you for pointing this out. I/We agree with this comment. Therefore, I/we have.[Explain what change you have made. Mention exactly where in the revised manuscript this change can be found – page number, paragraph, and line.]

“[updated text in the manuscript if necessary]”

Comments 2: [Paste the full reviewer comment here.]

Response 2: Agree. I/We have, accordingly, done/revised/changed/modified…..to emphasize this point. Discuss the changes made, providing the necessary explanation/clarification. Mention exactly where in the revised manuscript this change can be found – page number, paragraph, and line.]

“[updated text in the manuscript if necessary]”

4. Response to Comments on the Quality of English Language

Point 1:

Response 1:    (in red)

5. Additional clarifications

Reviewer 2 Report

Comments and Suggestions for Authors

I have some feedback on the reviewed manuscript:

Title Formatting: The title of the manuscript requires formatting improvements. There are inconsistencies in capitalization. Please ensure that the title follows a consistent capitalization style.

Participant Gender: It's noteworthy that the authors exclusively recruited female participants. However, the rationale for this gender-specific recruitment is not provided in the manuscript. The title should be revised to accurately reflect this gender-specific inclusion.

Abbreviation Clarification: In line 115, it's advisable to spell out "SE" as "standard error" when it is first introduced in the study. This will improve clarity for readers.

Sample Size Estimation: The manuscript does not mention whether a sample size estimation was conducted prior to the study. Adding this information would enhance the transparency of the research methodology.

IRB Approval: To ensure ethical compliance, the manuscript should specify the Institutional Review Board (IRB) approval number or provide relevant details regarding ethical approval.

Figure Presentation: Figure 2 should be presented as a bar chart instead of a line chart, as it does not represent longitudinal follow-up data. This change will offer a more suitable visualization of the data.

Author Response

For research article: Modulation of Motor Awareness: a Transcranial Magnetic Stimulation Study in the Healthy Brain

Response to Reviewer 2 Comments

1. Summary

2. Questions for General Evaluation

Reviewer’s Evaluation

Response and Revisions

Does the introduction provide sufficient background and include all relevant references?

Must be improved

We thank the reviewer for his/her comments. We have improved these points throughout the manuscript.

Are all the cited references relevant to the research?

Must be improved

Is the research design appropriate?

Must be improved

Are the methods adequately described?

Must be improved

Are the results clearly presented?

Must be improved

Are the conclusions supported by the results?

Must be improved

3. Point-by-point response to Comments and Suggestions for Authors

Comments 1: Title Formatting: The title of the manuscript requires formatting improvements. There are inconsistencies in capitalization. Please ensure that the title follows a consistent capitalization style.

Response 1: We thank the reviewer for his/her suggestion. We have revised the title as requested.

Comments 2:  Participant Gender: It's noteworthy that the authors exclusively recruited female participants. However, the rationale for this gender-specific recruitment is not provided in the manuscript. The title should be revised to accurately reflect this gender-specific inclusion.

Response 2: We thank the reviewer for his/her observation. Nevertheless, our sample (N=14) did not consist entirely of female participants, although they were the majority (11/14). We have clarified the gender composition in the Participants section (section: Materials and Methods, page: 3, line: 126).

Comments 3: Abbreviation Clarification: In line 115, it's advisable to spell out "SE" as "standard error" when it is first introduced in the study. This will improve clarity for readers.

Response 3: We apologize for the inconvenience, we have corrected the mistake in the manuscript (section: Materials and Methods, page: 3, line: 127).

Comments 4: Sample Size Estimation: The manuscript does not mention whether a sample size estimation was conducted prior to the study. Adding this information would enhance the transparency of the research methodology.

Response 4: We thank the reviewer for his/her observation . To decide our sample size, we referred on previous TMS studies on motor cognition (N=7, Haggard and Magno, 1999 Exp Brain Res; N= 10, Christensen et al., 2010, PLoS ONE; N= 16, Pyasik et al., 2019, Social cognitive and affective neuroscience) and/or cognitive studies using1 Hz rTMS (N= 15 per group, Ando’ et al., 2015, Brain Res; N= 13, e.g. Salatino et al., 2019, Front Psych). We have now added this point in the methodology section (section: Materials and Methods, page 3, line: 127-130).

Comments 5: IRB Approval: To ensure ethical compliance, the manuscript should specify the Institutional Review Board (IRB) approval number or provide relevant details regarding ethical approval.

Response 5: In the previous version of the manuscript we specified the IRB approval number in the dedicated section (before the references). We have now specified this information also in the metholodology section (section: Materials and Methods, page 3, line: 132-133).

Comments 6: Figure Presentation: Figure 2 should be presented as a bar chart instead of a line chart, as it does not represent longitudinal follow-up data. This change will offer a more suitable visualization of the data.

Response 6: We thank the reviewer for his/her observation. In order to facilitate the comparison of Figure 2 with Figure 3 (showing the significant interaction COND X SIDE X HAND), we decided to use a similar line chart.                   

Round 2

Reviewer 2 Report

Comments and Suggestions for Authors

The authors have revised their article well.